
# Evaluating High-Frequency radar data assimilation impact in coastal ocean operational modelling

Jaime Hernandez-Lasheras[1], Baptiste Mourre[1], Alejandro Orfila[2], Alex Santana[1], Emma Reyes[1], and Joaquín Tintoré[1,2]

[1]SOCIB - Balearic Islands Coastal Observing and Forecasting System. Palma, Mallorca, Spain
[2]IMEDEA (CSIC-UIB) - Instituto Mediterraneo de Estudios Avanzados. Esporles, Mallorca, Spain

**Correspondence:** Jaime Hernandez-Lasheras (jhernandez@socib.es)

**Abstract.**

The impact of the assimilation of HFR (High-Frequency Radar) observations in a high-resolution regional model is evaluated, focusing on the improvement of the mesoscale dynamics. The study area is the Ibiza Channel, located in the Western Mediterranean Sea. The resulting fields are tested against trajectories from 13 drifters. Six different assimilation experiments

are compared to a control run (no assimilation). The experiments consists in assimilating (i) Sea surface temperature, sea level anomaly and Argo profiles (generic observation dataset); the generic observation dataset plus (ii) HFR total velocities and (iii) HFR radial velocities. Moreover, for each dataset two different initialization methods are assessed: a) restarting directly from the analysis after the assimilation or b) using an intermediate initialization step applying a strong nudging towards the analysis fields. The experiments assimilating generic observations plus HFR total velocities with the direct restart provides the best

results, improving by 53% the average separation distance between drifters and virtual particles after the first 48 hours of simulation in comparison to the control run. When using the nudging initialization step, the best results are found when assimilating HFR radial velocities, with a reduction of the mean separation distance by around 48%. Results show the capability of the Ensemble Optimal Interpolation data-assimilative system to correct surface currents not only inside but also beyond the HFR coverage area. The assimilation of radial observations benefits from the smoothing effect associated with the application of the

intermediate nudging step.

## 1   Introduction

High-frequency radars (HFR) are a fast-growing technology, playing an important role in coastal observing systems around the world (Roarty et al., 2019). They allow real-time measurements providing a new, detailed, and quantitative description of physical processes at the marine surface (Paduan and Washburn, 2013). Their capacity to measure currents at high spatial

and temporal resolution over relatively large coastal areas make them a convenient system for operational purposes. They can be used to validate numerical models (Aguiar et al., 2019; Mourre et al., 2018; Lorente et al., 2021), analyze Lagrangian dynamics (Hernández-Carrasco et al., 2018) or constrain numerical models via data assimilation (Vandenbulcke et al., 2017; Iermano et al., 2016; Janeković et al., 2020).



HFR is a cost-effective shore-based remote-sensed technology exploiting the Bragg resonance phenomenon (Crombie, 1955)
to map ocean surface currents, wave fields, and increasingly winds, in coastal areas. They complement satellite altimeter observations which are limited to larger scales and suffer limitations when approaching the coast (Vignudelli et al., 2019; Pascual et al., 2013). The capability of HFR to give realistic observations of surface currents has been widely validated (Chapman et al., 1997; Emery et al., 2004; Paduan et al., 2006). Furthermore it has been used to validate geostrophic currents computed from along-track altimetry (Pascual et al., 2015) and to correct sea surface height (SSH) altimeter fields (Roesler et al., 2013).

Regional ocean models are invaluable tools for operational oceanography (Wilkin and Hunter, 2013; Onken et al., 2008; De Mey-Frémaux et al., 2019). However, they are inevitably affected by errors from multiple sources, especially in highly dynamic coastal areas where conditions tend to change rapidly. In such places, assimilation of observations such as the ones provided by HFR can help to constrain the model solution and improve the forecast. Assimilation of HFR data has been successfully applied in different regions around the globe, starting from the pioneer study conducted by Breivik (2001) using an
optimal interpolation (OI) scheme. Since then, many different studies have explored the performance of HFR for data assimilation in ocean circulation models, using different assimilation schemes, data types and techniques. Authors have employed high frequency (hourly) or filtered data depending on the focus of the study and the dynamical processes of interest. The use of whether radial or total observations has also been object of study and debate. While radial velocities theoretically provide a larger amount of information without any data processing, they are also noisier than reconstructed total observations, which
incorporate some spatial smoothing. Besides, radials have a wider coverage, providing data in areas only covered by one antenna and can be used even in case of failure of the other antennas. Both kinds of observations have been used with satisfactory results. To our knowledge, Shulman and Paduan (2009) is the only work evaluating the contribution of both radial and total velocities in the same experiment. Their results show the capacity of the system to improve surface currents and circulation down to 120m depth in areas covered by two or more antennas, for both kinds of data. Depending on the position of the mooring with
respect to the coverage of the antennas, their validation showed a varying complex correlation against mooring observations when using radial or total observations. Using observations from only one antenna, Shulman and Paduan (2009) found that results were extremely variable and highly dependent on the direction of the bearing with respect to the dominant flow.

Oke et al. (2002) used an OI scheme to assimilate low-pass filtered surface total velocity measurements from an HFR array to correct model circulation. They used a so-called TDAP (Time-Distributed Averaging Procedure) initialization method
after analysis, which progressively applies data assimilation increments, to preserve appropriate dynamical balances. This data assimilation approach resulted in an increase of the correlation between model and observations from 0.42 to 0.78.

More recently, hourly reconstructed total currents have also been employed using both sequential (Ren et al., 2016; Paduan and Shulman, 2004) and variational data-assimilation schemes such as 4D-Var (Zhang et al., 2010; Wilkin and Hunter, 2013; Yu et al., 2016). However, depending on the model set-up and the oceanic processes of interest, the use of hourly data may
not be the most appropriate, as for instance in Kerry et al. (2016), where radial speeds and angles are spatially averaged onto the model grid and a 24 h boxcar-averaging filter is used to remove tides and inertial oscillations that are not resolved by the model. Kerry et al. (2018) show that among all the assimilated observations, HFR were the ones which had the larger impact on the currents and the transport in the Eastern Australian Current.



The use of hourly data in sequential data assimilation schemes is not straightforward, due to the analysis frequency which is generally larger than one hour. An option is to use an extended state vector as in Barth et al. (2008), who employed an ensemble based Kalman Filter (KF) method using hourly radial observations in the West Florida Shelf. For the initialization Barth et al. (2008) implemented a spatial filter and averaged the ensemble fields in an attempt to remove spurious variability before it is introduced into the model. Barth et al. (2011) and Marmain et al. (2014) employed a similar approach, using all radial hourly observations available during the assimilation window and an extended state vector to correct the wind forcing fields and boundary conditions respectively in a similar way to variational methods. While Barth et al. (2011) showed that the correction had a positive impact on the reconstructed winds and the SST in the German Bight, Marmain et al. (2014) found an improvement in surface currents in the North-Western Mediterranean Sea, although with some degradation on the density fields and under surface currents. Stanev et al. (2015) also used hourly radial observations to correct tidal currents in the German Bight. In an operational context and based on a spatio-temporal optimal interpolation (STIO), Stanev et al. (2016) demonstrated that their system had a good skill to correct currents even beyond the HFR covered area.

A comparison of the impact of both time-filtered and unfiltered HFR currents (with respect to a model with and without tides) was done in Shulman and Paduan (2009), showing that the sub-tidal period velocity simulations were similarly improved through the assimilation of either low-pass-filtered surface currents or instantaneous (hourly) surface currents. More recently, Vandenbulcke et al. (2017) using different KF schemes, with an extended state vector, assimilated hourly radial velocities to correct inertial oscillations in a regional model of the Ligurian Sea. They show an important effect on the correction of inertial oscillations during the first 12 hours, when considering all hourly observations in a 48-hour time-window instead of using only the corresponding to one single hour.

In the present study we aim at evaluating the impact in coastal ocean operational modelling of the assimilation of both HFR total and radial velocities, also exploiting different initialization methods after analysis. Our focus is on the correction of mesoscale structures and larger scale circulation, rather than inertial oscillations or tidal currents.

The study area is the Ibiza Channel (IC) (Fig. 1), which is the passage between the oriental coast of Spain mainland and the island of Ibiza. It is a crucial area for understanding mixing and transport processes in the Northwestern Mediterranean Sea. Two different water masses interact in the IC: (i) a relatively salty water that has already recirculated in the Western Mediterranean flowing southward along the shelf as the Northern Current, and (ii) a branch of the Modified Atlantic waters transporting fresher waters originally entering through the strait of Gibraltar and flowing northward (Pinot et al., 1994, 1995) on its easternmost part. The dynamics, and the ecological and economical importance of the area have raised a specific interest in understanding the relevant ocean processes (Heslop et al., 2012; Balbín et al., 2014; Pinot et al., 2002; Hernández-Carrasco et al., 2018; Vargas-Yáñez et al., 2021). The analysis of repeated observations along a glider endurance line in the Ibiza Channel has revealed a high variability of meridional transports over time scales of days to weeks (Heslop et al., 2012). This high variability due to the interaction of multiple processes with different water masses over a complex topography make the operational forecasting particularly challenging. The anthropogenic pressure in the region makes it necessary to develop accurate tools for Search and Rescue, oil spill forecasting or larval dispersion to efficiently respond to emergencies and protect ecosystems.





Since 2012, the Balearic Island Coastal Observing and forecasting System (SOCIB, Tintoré et al. (2013)) operates a CODAR

HFR system that monitors the IC with two antennas measuring hourly surface currents (Tintoré et al., 2020). Lana et al. (2016) validated the IC HFR observations against current-meter, ADCP and surface Lagrangian drifters, showing a good agreement and the absence of significant mean error (hereafter referred as bias). A joint analysis of HFR observations and surface winds in terms of Empirical Orthogonal Functions (EOF) demonstrated that the surface current variability was mainly driven by local winds and mesoscale circulation.

Seven one-month period simulations have been generated to investigate the data assimilation performance of HFR raw radial observations compared to reconstructed totals currents. We have employed three different datasets, and for each of them, two different initialization methods after analysis. Additionally, a free-run simulation without assimilation has been used as control run. An exhaustive assessment has been performed following both Eulerian and Lagrangian approaches, including an independent set of 13 drifters deployed in the area.

The paper is structured as follows: Section 2 describes the data and methods employed, including the DA system and the description of the experiments. Results are presented in Section 3. Finally, the discussion of the results and the conclusions are presented in Sections 4 and 5.

## 2 Data and Methods

### 2.1 High Frequency Radar

The SOCIB HFR system consists in two CODAR SeaSonde stations of the islands of Ibiza and Formentera (named GALF and FORM, respectively), covering the eastern side of the IC. It operates since June 2012, providing real-time high-resolution observations of surface currents (Tintoré et al., 2020; Lana et al., 2015, 2016). Each HFR station emits at a central frequency of 13.5 MHz and a bandwidth of 90 kHz, reaching ranges up to 85 km. Emitted electromagnetic waves are back-scattered by surface waves of exactly half the HFR wavelength. Radial velocities (velocities toward or away from the antenna) are derived

from the Doppler shift due to the difference between ideal and measured Bragg frequency (Barrick, 2008). At the specified operating frequency, measurement depth is approximately 0.9 m (Stewart and Joy, 1974). Radial observations provide the velocity along a bearing, calculated from radio signals backscattered from the ocean surface. Hourly radial velocity maps from both stations are systematically quality controlled and the total velocity vectors are reconstructed by combining the radial velocities with overlapping coverage, on a regular $3 \times 3$ km grid. Each grid point observation is computed using a

unweighted least-square fitting (UWLS) (Lipa and Barrick, 1983), considering all radial observations within a 6km radius. Total reconstructed observations have a range up to 65 km off the antenna, compared to the 85 km that radials can reach.

In our experiments, we used daily means of raw radial observations and total currents (Figure 1) to filter the high frequency signals (i.e. tidal and inertial motions) and focus on the correction of the subtidal processes. Notice that tides in this area have very low amplitudes of the order of a few centimeters only. For the total currents the daily mean is only considered at grid points

where, during each day, at least 50% of hourly measurements are both available and flagged as good, as also used by Lorente et al. (2015). This threshold is more restrictive in the case of the radial observations. As mentioned, total derived currents at





each grid point are calculated using radial observations in a radius of 6km. It sometimes happens that there are enough radial observations to compute the total observation for most of the periods but with none of those radial observations satisfying the temporal threshold by itself. Using the same threshold, this would lead to patches with available reconstructed daily mean total

currents, but no daily mean radials available. Thus, we decided to reduce the temporal threshold for the radials to 25% (at least 6 hourly observations by day) to have better spatial coverage, consistent with that of the total observations in the area covered by both antennas.

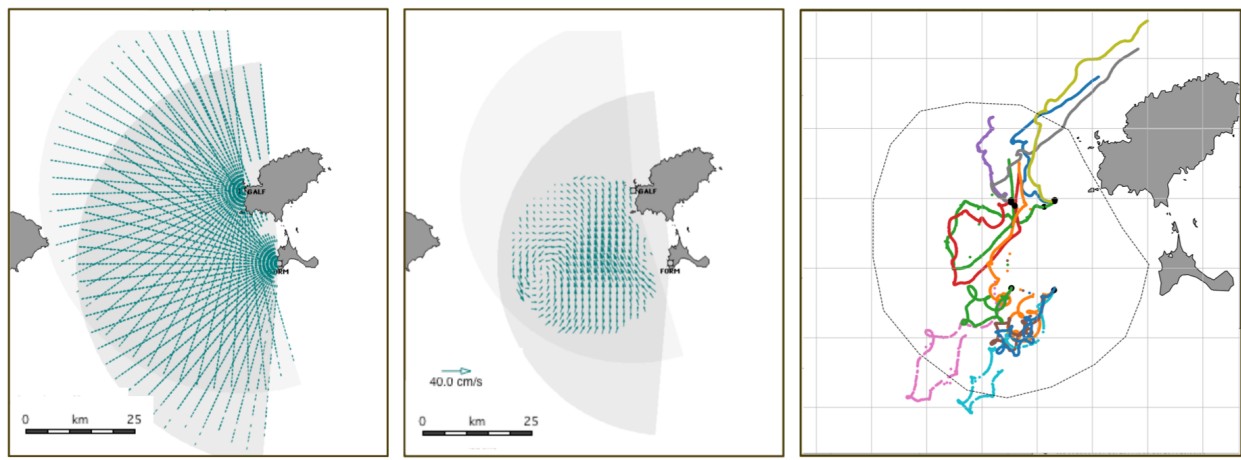

**Figure 1.** Map of the Ibiza Channel showing the HFR coverage area for radial (left panel) and total (central panel) currents, together with the position of the two antennas (GALF and FORM). The right panel shows the 13 drifters used for validation and their trajectories within the first 6 days after deployment. Each drifter has a randomly assigned color. Dots indicate start locations of the trajectories.

## 2.2 Regional model configuration

The Western Mediterranean OPerational system (WMOP, Juza et al. (2016); Mourre et al. (2018)) is a high-resolution regional

configuration of the ROMS (Regional Ocean Modelling System) model (Shchepetkin and McWilliams, 2005) for the western Mediterranean Sea. The spatial coverage spans from Gibraltar strait on the West to the Sardinia Channel on the East (6ºW-9ºE, 35ºN-44.5ºN, see Fig. 1) with a horizontal resolution around 2 km and 32 vertical sigma levels (resulting in a vertical resolution between 1 and 2m at the surface). The WMOP system is used to produce daily forecasts of the regional ocean circulation, which is used for a wide range of applications including search-and-rescue and analysis of plastic, parasite or larval dispersion for

instance (Calò et al., 2018; Ruiz-Orejón et al., 2019; Cabanellas-Reboredo et al., 2019; Compa et al., 2020; Torrado et al., 2021; Kersting et al., 2020).

The vertical mixing coefficients are set using the Generic Length Scale (GLS) turbulence closure scheme (Umlauf and Burchard (2003), with parameters p=2.0; m=1.0; n=-0.67 as in line 1 of their Table 7). The bathymetry is derived from a 1' database (Smith and Sandwell, 1997). The simulation used in this study is initialized from and nested within the larger



scale Copernicus Forecasting System (CMEMS MED-MFC), with a 1/16º horizontal resolution (Simoncelli S., 2017). The atmospheric forcing is provided every 3 hours at $1/20°$ resolution by the Spanish Meteorological Agency (AEMET) through the HIRLAM model (Undén et al., 2002). These fields are used to compute surface turbulent and momentum fluxes through bulk formulae. Atmospheric pressure forcing is neglected to avoid SSH high-frequency variability issues. Inflows from the six major rivers in the region are considered as point sources, using daily climatological values. Tides are not considered in the

model.

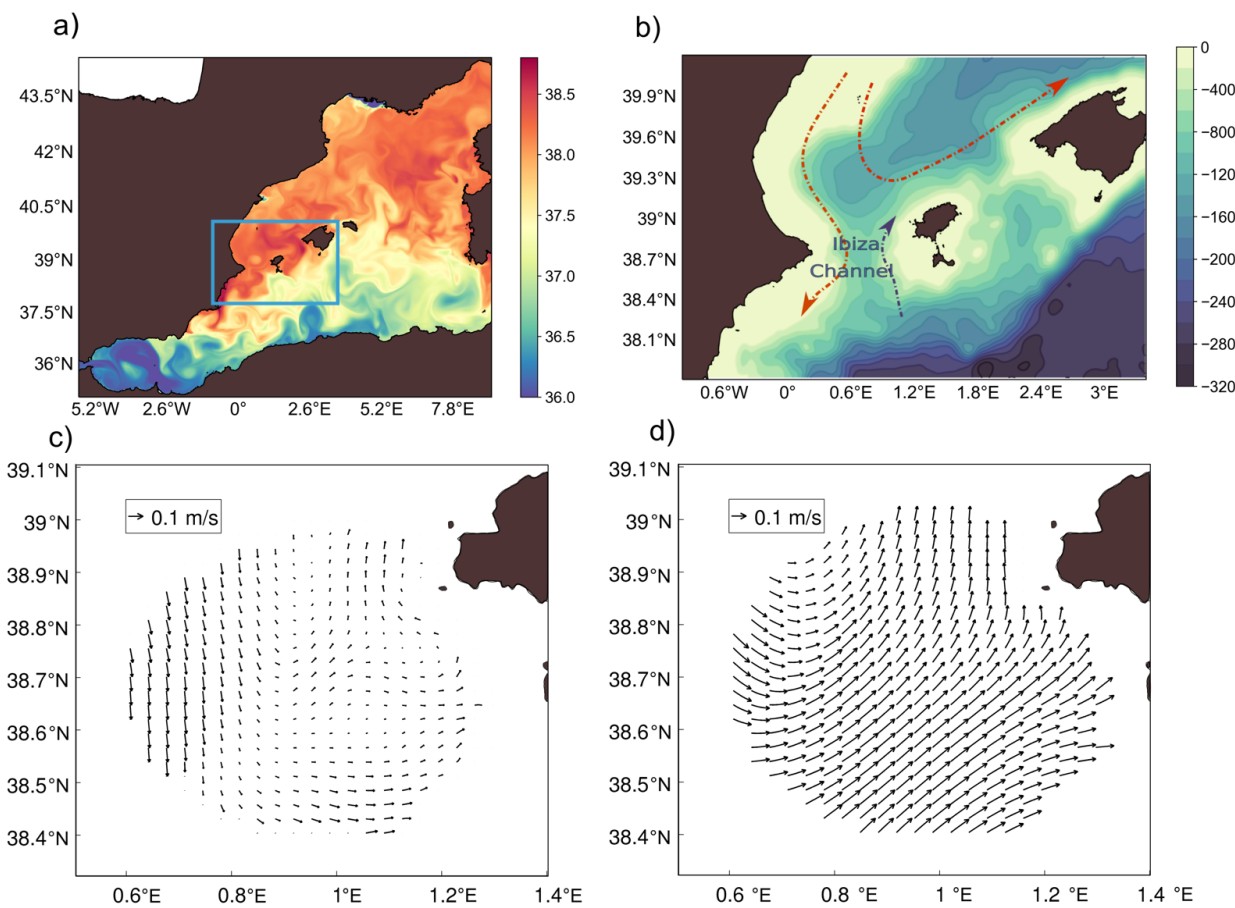

**Figure 2.** Illustration of the modelling domain and study area. a) WMOP sea surface salinity. The Ibiza and Mallorca Channel area is delimited by the blue rectangle. b) Bathymetry and main circulation features in the Ibiza Channel. c) Mean HFR surface currents over the whole simulation period (20 September to 20 October 2014). d) Mean surface field currents over the whole simulation period computed from the model.

A multi-year free-run hindcast spanning the period from 2009 to 2018 (Mourre et al., 2018; Aguiar et al., 2019) has been used as control simulation. This simulation also provides the initial state for the data assimilative simulations starting on 20



September 2014. Fig 2 shows the mean surface field of the control run during the simulation period (20 September to 20 October 2014) together with the mean surface currents measured by the HFR for the same period. The HFR observations

depict an average southward current west of 0.8E. This current is deviated towards the south-east of 38.7N, and the flow is directed northward in the eastern side of the coverage area, close to Ibiza and Formentera coast. The control run represents this overall pattern, but with a significant overestimation of the mean velocities and a spatial mismatch of the eastward deviation of the flow (this deviation occurs too much to the east in the model).

## 2.3  Data Assimilation System

The assimilation scheme employed here is the multimodel local Ensemble Optimal Interpolation (EnOI) employed in Hernández-Lasheras and Mourre (2018). It is a form of the EnOI, which has been a widely used scheme, since it represents a cost-effective alternative compared with more complex methods as the Ensemble Kalman Filter or the 4Dvar (Oke et al., 2002; Evensen, 2003; Counillon and Bertino, 2009). EnOI is a 3D sequential assimilation method that allows the use of a large ensemble size together with localization. A stationary ensemble of model simulations is used to estimate background error covariances.

The WMOP-DA system consists of a sequence of analyses (model updates given a set of observations) and model forward simulations.

For each analysis, the state vector $\mathbf{x} = (\mathbf{T}_{i,j,k}, \mathbf{S}_{i,j,k}, \mathbf{u}_{i,j,k}, \mathbf{v}_{i,j,k}, \mathbf{SSH}_{i,j})^{\mathbf{T}}$, contains the model trajectory, i.e., the prognostic model variables at all wet gridpoints i, j, k.

During the analysis step, the state vector $\mathbf{x}^a$ is updated according to Eq. (1), where $\mathbf{x}^f$ is the background model state vector,

$\mathbf{H}$ is the linear observation operator projecting the model state onto the observation space and $\tilde{\mathbf{K}}$ is the Kalman gain estimated from the sample covariances (Eq. 2). $\mathbf{y}$ is the vector of observations. Matrices $\tilde{\mathbf{P}}^f$ and $\mathbf{R}$ are the error covariance matrices of the model and the observations, respectively.

$$\mathbf{x}^a = \mathbf{x}^f + \tilde{\mathbf{K}}(\mathbf{y} - \mathbf{H}\mathbf{x}^f), \tag{1}$$

$$\tilde{\mathbf{K}} = \tilde{\mathbf{P}}^f \mathbf{H}^T (\mathbf{H}\tilde{\mathbf{P}}^f \mathbf{H}^T + \mathbf{R})^{-1}, \tag{2}$$

$\tilde{\mathbf{P}}^f$ contains the background error covariances (BECs). In our approach we estimated the BECs by sampling three long-run simulations of the WMOP with different initial and boundary forcing provided by the CMEMS MED-MFC (Simoncelli S., 2017) and CMEMS GLO-MFC (Lellouche et al., 2018) forecasting systems and varying momentum and diffusion parameters. An ensemble of 80 realizations is considered in each analysis. Each ensemble member is randomly extracted from the three

different long-run simulations within a temporal window of 90 days centered on the day of the analysis for the different years covered by the three long-run simulations. The seasonal cycle is removed from the multivariate fields before computing the ensemble anomalies to limit the effects of large scale correlations, mainly in terms of surface temperature. This way, we obtain multivariate, inhomogeneous and anisotropic 3-dimensional model BECs characteristic of the mesoscale variability. We used a domain localization of 200 km, corresponding to the average distance between two Argo profiles in the Western Mediterranean





Sea. An independent analysis is performed for each water column of the model domain, considering only the observations within the localization radius.

    We used a diagonal observation covariance error matrix $\mathbf{R}$. The observation error for all HFR observations has been considered the same, with a value of $0.1\mathrm{m/s}$ (accounting for instrumental and representativity errors). Note that Lana et al. (2016) state that the maximum instrumental error was about $0.04\mathrm{m/s}$. Besides, radial observations were synthetically generated from

the total ones, and the result of the assimilation was compared to that of the assimilation of the corresponding total velocities, obtaining similar results. The state vector equivalents of HFR radials are obtained using the following equation:

$$\mathbf{H}\mathbf{x}^{f} = u_{x}\cos\alpha + u_{y}\sin\alpha, \tag{3}$$

where $u_{x}$ and $u_{y}$ are the model surface velocity components interpolated at the observation point, and $\alpha$ denotes the angle (anti-clockwise towards the east) pointing from an antenna station to a certain location.

A 3-day assimilation cycle is applied with different time windows for each source of observation as explained in the following section. In each analysis (*day n*) the daily average field is employed as background and two different initialization approaches (Fig. 3) have been applied to restart the model after the analysis. Sequential assimilation methods are affected by initialization issues, as primitive equation models are sensitive to discontinuous changes in their model fields (Oke et al., 2002). These discontinuities may introduce artificial waves or structures in the model that affect the quality of predictions. Different strategies

have been proposed to address this problem (Sandery et al., 2011; Yan et al., 2014).

    The simulation for *day n+1* restarts directly from the results of the analysis. The second approach, which will be referred to as *nudging* consists in running again the *day n* applying a very strong nudging (time scale of one day) towards the temperature, salinity and SSH fields provided by the analysis. Notice that the nudging is not applied to the velocity fields. These are adjusted by the model itself according to its dynamics. This procedure reduces the model corrections but guarantees updated multivariate

fields closer to model equation balances, which limits instabilities. This set-up is very similar to the one employed by Oke et al. (2007) in the Bluelink system.

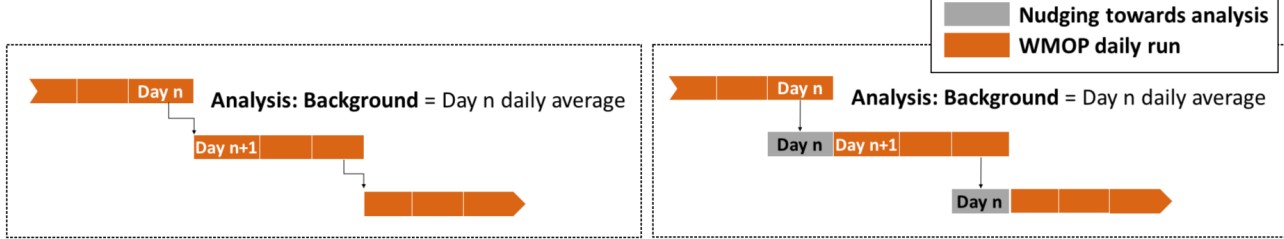

**Figure 3.** Data Assimilation procedure, illustrating the two initialization methods and the 3-day cycles. The diagram on the left describes the direct initialization strategy from the analysis. The diagram on the right describes the *Nudging* strategy for initialization. Orange rectangles represent each 1-day run of WMOP. Grey rectangles represent the 1-day run of WMOP in which a strong nudging towards the results of the analysis is applied.





## 2.4 Simulations

Seven simulations of WMOP are used to investigate the impact of both HFR observation and initialization methods (Table 1). The period selected for the simulation experiments covers one month, from September $20^{\text{th}}$ to October $20^{\text{th}}$ 2014, assimilating

different sets of observations every 3 days. During this period a total of 13 satellite-tracked surface drifters (Tintoré et al., 2014) were deployed in the area covered by the HFR and used as independent data for validating the numerical experiments (Fig. 1). We adopted the operational prediction setup of WMOP, considering only observations before the analysis date. Notice that a "retrospective analysis" framework considering a time window centered on the analysis date could slightly improve the results presented in this paper. However, since our objective is to implement this method for daily predictions, the operational setup

has been selected. Satellite SLA (sea level anomalies), SST (sea surface temperature) and T-S (temperature and salinity) Argo profiles, defined as the Generic Observing sources (GO), are assimilated in all these simulations. The SLA consists in along-track L3 multi-satellite reprocessed observations provided by CMEMS. We consider a 3-day window for SLA observations. The SST comes from a L4-GHRSST foundation SST product distributed by JPL-MUR (NASA/JPL, 2015). The foundation SST is the temperature free of diurnal temperature variability, corresponding to the temperature of the surface just before

the daily heating by the sun. Since the model daily average contains the signature of the diurnal cycle, this effect needs to be accounted for in the representativity error. This is approximated by computing the variance of the difference between the model SST field at 8 a.m. and the daily average field used as background for each of the grid points. The ultra-high 1 km resolution gridded fields have been smoothed and interpolated to a 10 km grid to limit the number of observations, while still representing the effective scale that this SST product can resolve (Chin et al., 2017). For the T-S Argo profiles we have

considered a 5-day time-window, which corresponds to the nominal time of Mediterranean Argo floats cycles. For each profile, values are binned vertically to obtain a single value for each model grid cell. The variance of the data within a bin is used as the vertical representation error, which is added to the horizontal one, assumed to be $0.25°\text{C}^2$ and $0.05^2$ for temperature and salinity measurements, respectively.

A control run ($CR$) without data assimilation has been used as benchmark to assess the performance of the different assim-

ilation experiments. We called $GNR$ the simulation in which we only assimilated GO. Additionally, four other simulations assimilating HFR data together with GO have been generated. In all four cases we assimilate daily averages to remove the impact of inertial oscillations and tides, which are not the focus of this study. Daily averaged fields from the model are used as background for the analysis. $TOT$ simulation employs HFR totals, computed as described before. We called $RAD$ the simulation assimilating all possible daily mean radial observations.

Data assimilation experiments have been repeated using both types of initialization for every dataset. Our analysis will first evaluate the impact and trade-offs of the different kind of HFR observations when using the direct restart from the analysis procedure. Then, the impact of the nudging initialization method will be specifically discussed.





| Experiment | Assimilated observations |
|:---:|:---:|
| **CR** | None |
| **GNR** | SLA, SST, TS |
| **TOT** | SLA, SST, TS, HFR totals |
| **RAD** | SLA, SST, TS, HFR radials |

**Table 1.** Basic description of the experiments, indicating the dataset used in the simulations.

## 3 Results

### 3.1 Assessment of the impact of DA on SST, SLA and T-S profiles over the whole domain

To evaluate the general performance of the DA, all one-month-long experiments are first compared against SLA, SST and Argo TS profiles over the whole modelling domain. For each experiment, the WMOP fields are interpolated onto the observation points.

Each day, the model SLA has been assessed against along-track daily multi-satellite altimetry observations provided by CMEMS. Model daily mean fields at the observation points are considered. The satellite SST product is compared to the
model SST at 8 a.m. to reduce the potential impact of the diurnal cycle. For the comparison of the model and the Argo T-S profiles, the available daily observations are compared against model daily mean fields. The closest grid point of the model has been considered. Due to the backward-in-time assimilation window, the observations used for the validation have not been previously assimilated. However, they can not be considered as fully independent since the data employed for the validation come from the same platforms that provide the assimilated measurements.

Taylor diagrams (Taylor, 2001) are presented here for the evaluation of the simulations. They illustrate the correspondence between model and observations in terms of correlation coefficient, centered root mean square difference (CRMSD) and standard deviation. However, note that the diagram does not represent the mean error between the observations and the model, which has been examined separately. The magnitude of the SST mean error decreases from -0.29°C to -0.14°C , representing in all simulations less than the 14% of the total RMSD [1]. The mean error between the $CR$ and the Argo profiles is 0.4°C and
-0.13 for temperature and salinity respectively, representing less than 8% of the RMSD in both cases.

The use of DA results in a significant improvement of both the SLA and SST fields, as shown in Fig. 4. For both data sources the symbols corresponding to each simulation assimilating data overlap, meaning that the validation metrics are very similar for all of them. For the SLA it leads to a significant increase in the correlation, with values from 0.42 to around 0.70, and a 30% reduction in the CRMSD for all the experiments with DA. Notice that the model SLA presents a relatively large mean
error, with a value of around -0.07m. Discrepancies are common when comparing models to altimetry due to differences in the mean sea level. This mean error, which persists after DA, accounts for the difference between the mean dynamic topography

---

[1]The RMSD has a contribution from the bias and CRMSD according to the following formula: $RMSD^2 = CRMSD^2 + bias^2$





of the model and observations. This way, the reduction in the RMSD is mostly due to reductions of the CRMSD, which can be observed in the diagrams.

Concerning the SST, we obtain a similar error reduction in terms of centered RMSD, of the order of 30% closer to observations when using DA. An increase in correlation is also obtained, from 0.82 to around 0.92 when compared with the CR. We do not observe a significant difference between the simulations using different datasets.

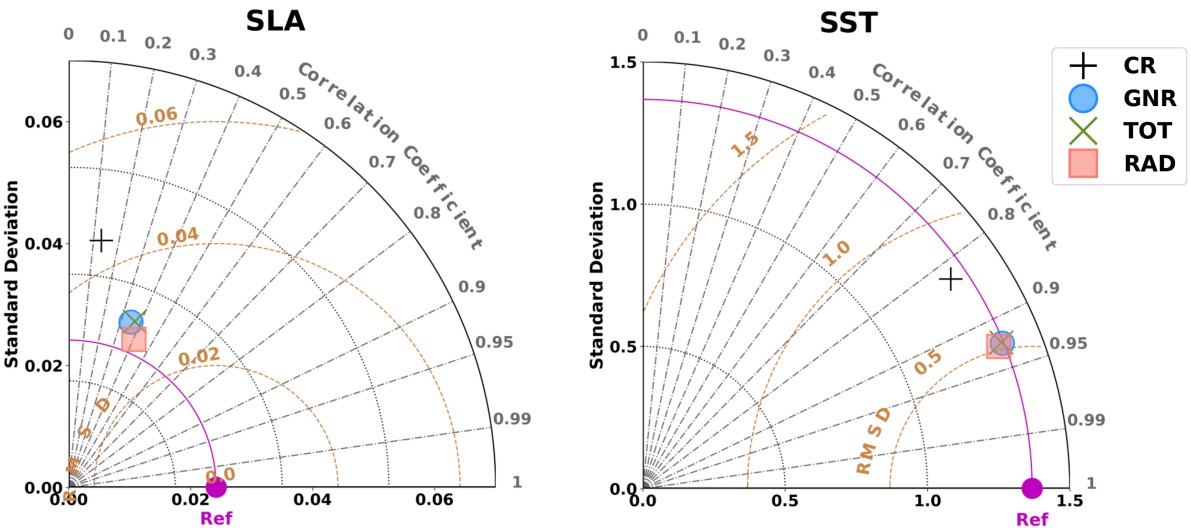

**Figure 4.** Taylor diagrams comparing models and observations in terms of SLA (left) and SST (right) over the whole modelling domain. X and Y axis represent the standard deviations of the data. Distance from the reference point located on the X axis ( noted as Ref. in magenta) represents the centered root mean square deviation (CRMSD). Correlation between observations and model increases clockwise. Symbols represent the different simulations, as specified in the legend

Similar conclusions are obtained when examining the Taylor diagrams focusing on Argo temperature and salinity profiles (Fig. 5). Although the CR simulation shows a very high correlation with observations (0.88 and 0.95 for temperature and salinity respectively), this correlation is further increased for the experiments with DA. A CRMSD reduction of more than 35% is obtained for both salinity and temperature observations in all data assimilative experiments. The diagram for the salinity shows a decrease in the standard deviation with DA and slight differences between $RAD$ and the other two simulations.

The impact of the assimilation on the different fields has been also evaluated considering only observations surrounding the IC area, leading to similar results.

## 3.2 Eulerian assessment of the impact of DA on surface currents

To evaluate the DA capabilities in improving the representation of surface currents, we performed an Eulerian analysis in the HFR coverage area. WMOP surface daily mean velocities are compared against HFR totals daily mean fields. The total



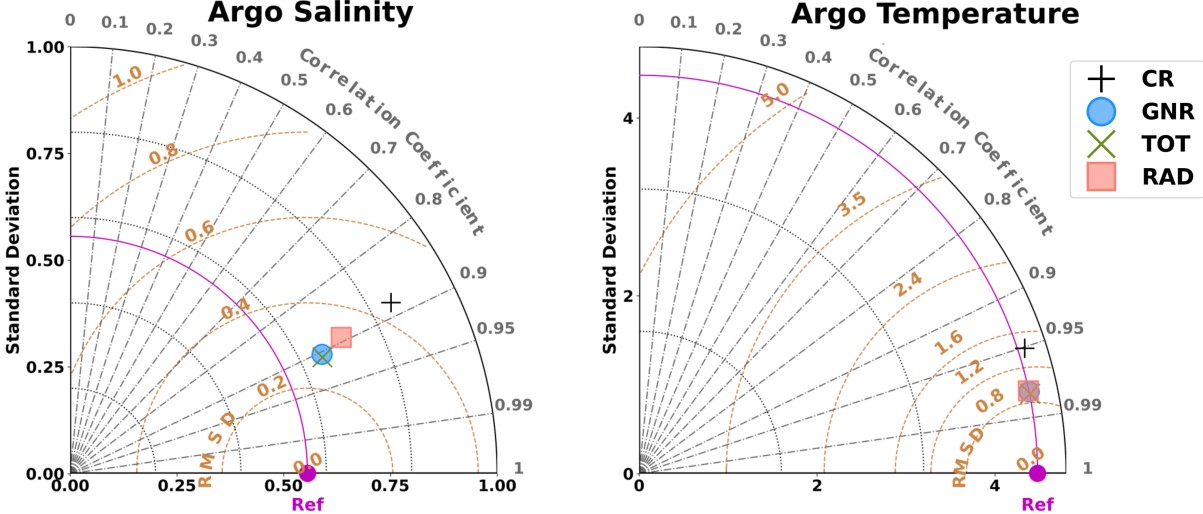

**Figure 5.** Same as Figure 4 for Argo temperature (left) and salinity (right) profiles.

observations are derived from the radial data, as described in section 2.1. The daily mean field is only computed at those points that provide more than 50% of hourly data. The model is then interpolated to HFR observation grid points. As for the SLA, SST and Argo TS profiles, the validation can not be considered here as fully independent, since we are using the same observing
platform. However, the data used for validation at a given time have not yet been assimilated in the model.

The performance in terms of surface currents is first analyzed by using the Taylor diagrams for the velocity components (Fig.6). Observing the zonal velocity component, it experiences a strong correction with the assimilation of GO. Specifically, the CRMSD suffers a reduction of 28% while the correlation increases towards the observation from 0.28 to 0.44. This performance is further improved by the two experiments using HFR data, with more than a 40% reduction in CRMSD. While $TOT$
experiment exhibits the largest error reduction, $RAD$ provides the best correlation with observations (0.7), compared to 0.63 obtained by $TOT$.

Considering the meridional velocity component, we can observe how $GNR$ has a lower correlation and higher CRMSD than the $CR$. Here, the use of HFR observations is necessary to reduce the difference between model and observations. The correlation slightly increases, with the best results obtained for $TOT$ (0.47) and $RAD$ (0.43). Moreover, the standard deviation
and CRMSD display a significant reduction (27% for $TOT$ and 19% with the radial observations).

Figure 7 shows the spatial distribution of the surface current speed mean error (bias), defined as the difference between the HFR and the model in terms of the module of the velocities at each grid point. Positive error reflects that observation mean values are larger than model estimates. The control run, $CR$, overestimates the currents in most of the domain with the exception of a small area on the western side. The mean bias over the whole HFR domain is of 10.4 cm/s (see Table 2). DA
corrects the bias in all 3 experiments. The assimilation of GO leads to a reduction of the error over the whole domain, with a

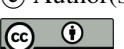



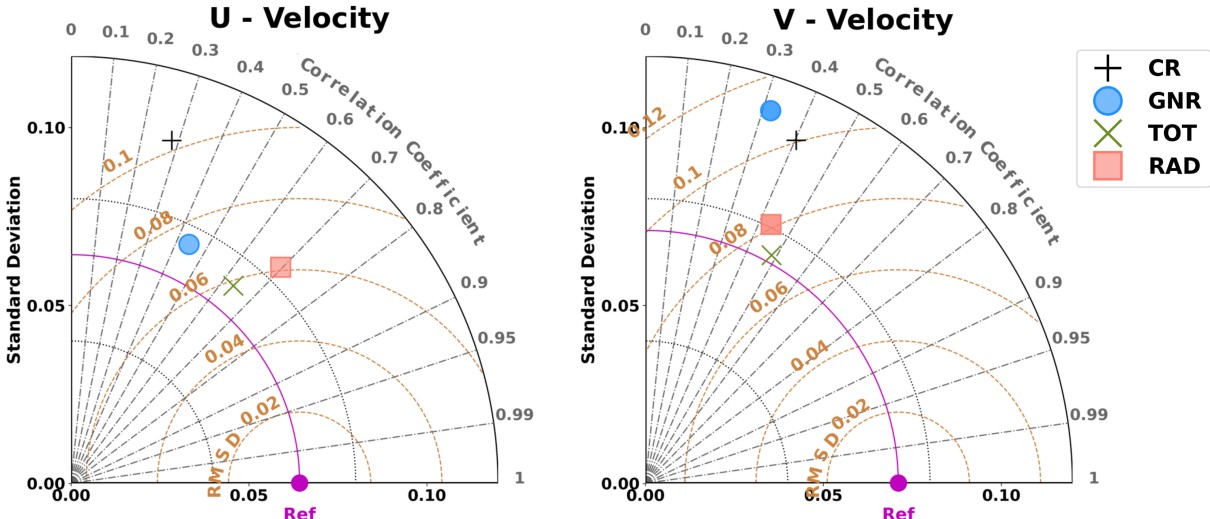

**Figure 6.** Taylor diagrams for WMOP simulations compared to HFR surface currents observations. Separate diagrams for each velocity component: U (left) and V (right). The symbols represent the different simulations, as specified in the legend

mean value of 5.4 cm/s. A further reduction is achieved when assimilating HFR velocities. $RAD$ has a mean bias of 2.6 cm/s, which is particularly higher near the Ibiza antenna, while $TOT$ has the lowest bias, with a mean value of 1.3 cm/s.

|  | Bias (cm/s) | nRMSD |
|---|---|---|
| **CR** | 10.4 | 1.0 |
| **GNR** | 5.3 | 0.79 |
| **TOT** | 1.3 | 0.52 |
| **RAD** | 2.6 | 0.57 |

**Table 2.** Bias and normalized RMSD between model surface currents speed and HFR's.

## 3.3 Lagrangian assessment of the impact of DA on surface currents

As previously stated, 13 surface drifters were deployed in the HFR coverage area during the simulated period, as decribed in
(Lana et al., 2016). Three different kinds of surface drifters (ODi, MDOi and CODEi) were employed, all drifting at a depth between the surface and 1m. No significant wind drag is expected for these drifters (more details can be found in Révelard et al. (2021) or Barth et al. (2021)). Virtual particle drifts were then computed using model surface currents. For each experiment, and for eight consecutive days (from October $1^{st}$ to October $8^{th}$), 1000 neutrally buoyant particles were launched at each of the positions of the 13 drifters at 00:00 on these dates. Lagrangian tracks were simulated using Ocean Parcels (Lange and



**Figure 7.** Mean error (bias) field of the total current speed. Mean speed is subtracted to observations at each grid point. Positive values indicate the model overestimates observations in average.

Van Sebille, 2017) and 5 days period of WMOP velocity fields (at 3-hours resolution). Additionally, we inserted a diffusion





term using a Brownian motion scheme with the objective of representing the impact of the subgrid processes not resolved by the model.

The distance between real drifters and the center of mass of each set of the 1000 Lagrangian particles is computed at each time step and a skill score (Eq. 4 and 5) is given for each drifter every day following the description made by Liu and Weisberg

(2011). A non-dimensional index $s$ is calculated based in the normalize cumulative Lagrangian separation distance, from purely Lagrangian parameters (Eq. 4), where $d_i$ is the separation distance between the modeled and observed endpoints of the Lagrangian trajectory at time step $i$ after initialization, $l_{oi}$ is the length of the observed trajectory, and $N$ is the total number of time steps.

$$s = \sum_{i=1}^{N} d_i \Big/ \sum_{i=1}^{N} l_{oi} \qquad (4)$$

$ss = 1 - s$ (5)

Trajectories are also simulated using the hourly DIVAnd reconstructed fields presented in Barth et al. (2021). DIVAnd is a n-dimensional variational analysis method which is used here to reconstruct hourly 2D vectorial fields from radial observations. It was shown to improve the reconstruction compared to the Open-boundary Modal Analysis (Kaplan and Lekien, 2007). Figure 8 provides a few examples illustrating the Lagrangian prediction capacity for the different simulations. Each panel shows

the trajectories of the drifter and the center of mass of the virtual particles for each experiment for 48 hours of simulation. These plots illustrate the diversity of situations associated with the spatio-temporal variability of the surface ocean velocities. In particular for panels b), c), d), and e), $TOT$ displays a very good agreement with the observations, resulting in the best performance overall. However, it is worth noting that this behavior is not systematic and the simulations assimilating HFR sometimes fail in providing the best trajectories (panels $a)$ and $f)$ ).

Figure 9 shows the skill score for all experiments (4 model simulations + DIVAnd fields) for a forecast horizon of 48 h. Each point is located at the initial position of the particles at the beginning of the Lagrangian simulations and represents the value of the skill score of the center of mass of the cloud of virtual particles. For trajectories with a cumulative separation distance larger than the cumulative distance traveled by the particles, the model has a negative skill score. On the other hand, values close to 1 indicate a nearly perfect match between the drifter and virtual trajectories. Values of the mean skill score are given in

table 3, where the mean skill score is computed separately for (i) the whole trajectories; (ii) the trajectories starting inside the HFR coverage area and (iii) trajectories whose initial position is outside the HFR area. The top left panel of Figure 9 shows the spatial distribution of the $SS$ for the $CR$, highlighting multiple trajectories inside the HFR coverage area for which the model has no skill ($SS < 0$) according to Liu and Weisberg (2011) criteria, resulting in a negative mean value (-0.35), and trajectories outside for which the model has some skill, with scores over 0.5, resulting in a mean values of 0.36. All experiments present

a larger $SS$ outside the HFR coverage area. This in mainly due to the characteristics of the trajectories north of the island of Ibiza, where the circulation is dominated by the Balearic current, with a more steady northeastward flow, generally reproduced in the model, as previously described by (Révelard et al., 2021). .


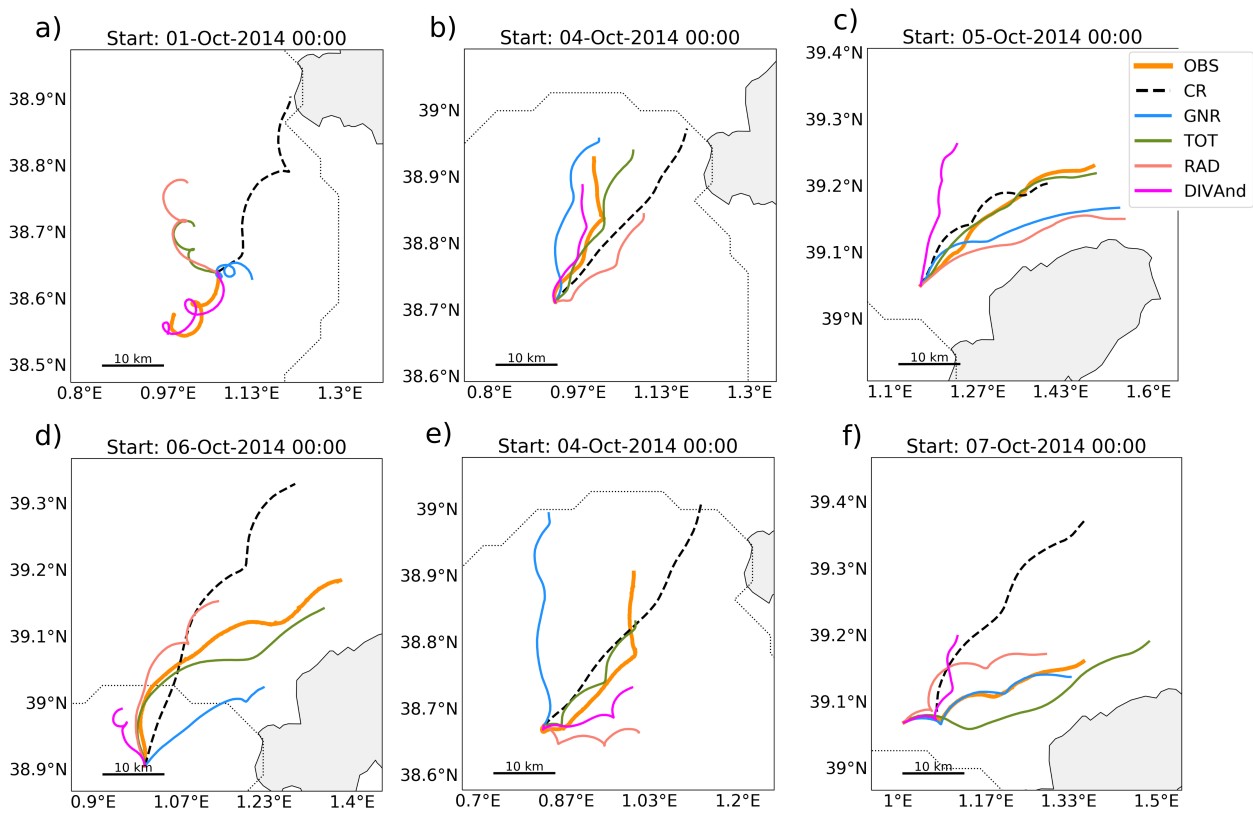

**Figure 8.** Map showing two-day satellite-tracked drifters and model trajectories derived from different DA experiments and corresponding to different dates. Model trajectories represent the trajectories of the center of mass of the 100 particles launched at the drifter position at 00:00 of the indicated starting day.

| Experiment | SS Whole domain | SS Inside | SS Outside |
|:---:|:---:|:---:|:---:|
| **CR** | -0.16 | -0.35 | 0.36 |
| **DIVAnd** | 0.45 | 0.51 | 0.28 |
| **GNR** | 0.19 | 0.05 | 0.58 |
| **TOT** | 0.45 | 0.41 | 0.57 |
| **RAD** | 0.42 | 0.36 | 0.59 |

**Table 3.** Average Skill Score for the different experiments, over the whole domain, and inside and outside the HFR coverage area.

In general, all experiments with DA improve the trajectories. In particular, $GNR$ increases the skill score compared to $CR$, resulting in mean values of 0.19 and -0.16, respectively, for both simulations over the whole domain. Note, however, that inside the coverage area not all trajectories are properly represented by the model. The assimilation of HFR data along with

GO further increases the skill score. The improvement is particularly significant inside the HFR domain, where most of the trajectories have positive $SS$. $TOT$ has the best results among the model experiments, with a mean value of 0.41 inside the coverage area, which in comparison to $RAD$ (0.36) is a clear improvement. Outside of the coverage area all data assimilative simulations lead to similar results, with $RAD$ obtaining the best score (0.59) among all. Assimilation of HFR can increase the

performance of the model outside the coverage area or in areas with observations from a single antenna (for the case of $RAD$), as in the area North-West of Ibiza, as shown in Figure 9.

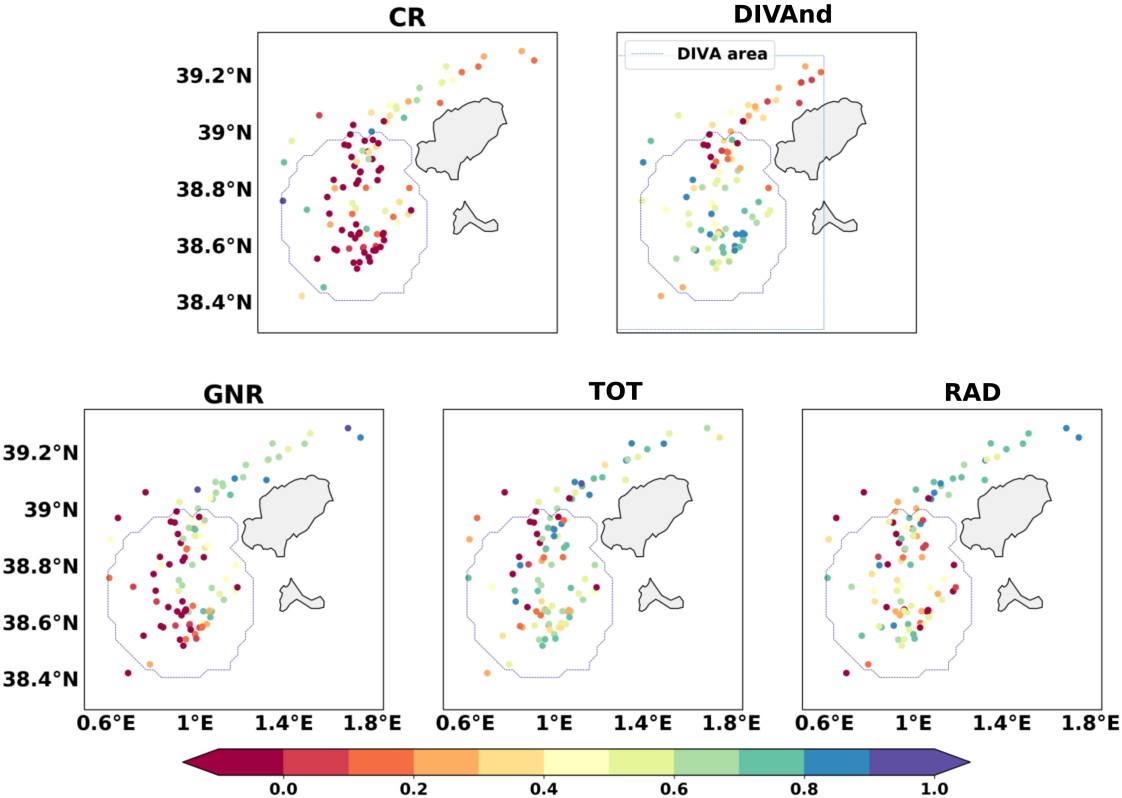

**Figure 9.** Scattered dots represent the skill score (Liu and Weisberg, 2011) of each simulation to represent a drifter trajectory. The dot position is the starting point of each Lagrangian simulation. Values lower than 0 mean the simulation has no skill at representing that certain drifter trajectories according to the metric used, while values close to 1 mean a perfect performance of the model. Skill score has been calculated for 48 hours.

The average separation distance is computed according to Equation 6, where $n_{drif} = 13$ is the number of drifters, $n_{part} = 1000$ is the number of particles and $\mathbf{x^d}$ and $\mathbf{x^v}$ are the positions of the real drifter and the corresponding virtual particle respectively (Figure 10). For each 5 day trajectory, the mean separation distance is first computed averaging over the number





of drifters, providing a single distance as a function of time $d(t)$ for the 13 drifters (Eq. 6). Then, the four values of $d(t)$, one
for each of the four simulations starting in consecutive days are averaged.

$$d(t) = \frac{1}{n_{drif}} \sum_{i=1}^{n_{drif}} \left( \frac{1}{n_{part}} \sum_{j=1}^{n_{part}} \left| \mathbf{x_i}^{\mathbf{d}}(\mathbf{t}) - \mathbf{x_{ij}}^{\mathbf{v}}(\mathbf{t}) \right| \right) \qquad (6)$$

The mean distance between virtual and real drifters is significantly reduced when DA is applied. The assimilation of GO
efficiently helps to reduce the mean separation distance, with a reduction of 31% after 48 hours compared to $CR$ (18.9 versus
27.2 km). Consistent with the previous analysis, the assimilation of HFR total observations along with the GO further increases
the performance, leading to the lowest mean separation distance (12.8 km), with a 53% reduction compared to the $CR$. The
use of radial observations also leads to a high reduction of the mean separation distance (48%), which is reduced to 14.3 km
after 48hr.

DIVAnd simulations present a mean distance of 8.4 and 17.3 km after 24 and 48 h respectively, affected by a significant
number of trajectories outside of the HFR coverage area and so, not properly constrained by DIVAnd.

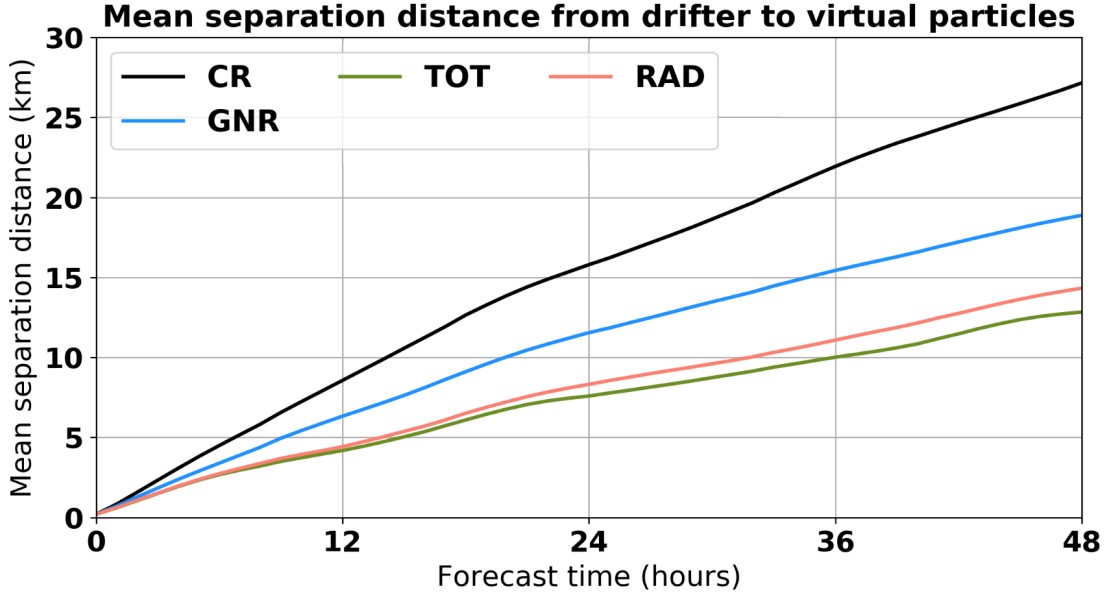

**Figure 10.** Mean separation distance between drifter and the center of mass of virtual particles using the direct restart from analysis


### 3.4 Impact of the nudging restart strategy

Overall, the results in the whole domain comparing to satellite and Argo observations are similar to those obtained for the
simulations restarting directly from the analysis. The improvement is slightly lower due to the nudging step, but all data





| Experiment | Bias (cm/s) | nRMSD |
|:---:|:---:|:---:|
| **CR** | 10.4 | 1.0 |
| **GNR-N** | 5.0 | 0.75 |
| **TOT-N** | 2.2 | 0.59 |
| **RAD-N** | 2.1 | 0.58 |

**Table 4.** Bias against surface current speed HFR observations and RMSD normalized with the CR error for the different simulations using nudging restart approach after DA.

assimilative simulations provide comparable metrics. The reduction of the RMSD compared to the $CR$ is around 8% for the
SLA, while for the SST is reduced around 30%. Considering Argo profiles, the reduction of the RMSD is of 35% for all simulations, both for temperature and salinity.

Table 4 presents the bias and normalized RMSD for total velocity. This has to be compared with Table 2, which shows the results for the previous simulations. We can observe a slight improvement for the $GNR - N$ simulation when using the nudging initialization in comparison to restarting directly from the analysis, with a reduction of both the bias and the RMSD.
While for the $RAD$ this initialization method also helps to reduce the bias compared to direct restart from the analysis, this is not the case when using total observations.

The Lagrangian assessment confirms these results, reflecting the usefulness of HFR data to correct surface currents using this initialization method even when the nudging is only performed towards the SSH and TS fields. The SS for the $GNR - N$ simulation increases significantly inside the coverage area while decreasing outside, with an average value of 0.39, larger than
the value of 0.34 obtained with the other approach.

The correction obtained using HFR Total velocities together with GO is slightly degraded using the nudging approach both inside and outside the coverage area, with an average $SS$ of 0.45. On the other hand, $RAD$ has a better $SS$ when using the nudging approach. The average $SS$ inside the radar domain increases from 0.38 to 0.43, while outside the domain it slightly decreases from 0.59 to 0.58.

| Experiment | SS Whole domain | SS Inside | SS Outside |
|:---:|:---:|:---:|:---:|
| **CR** | -0.16 | -0.35 | 0.36 |
| **GNR-N** | 0.28 | 0.18 | 0.54 |
| **TOT-N** | 0.41 | 0.35 | 0.56 |
| **RAD-N** | 0.43 | 0.38 | 0.57 |

**Table 5.** Skill Score mean for the different experiments using the nudging initialization method, overall mean, inside and outside the HFR coverage area.

The mean separation distance after 48h when assimilating GO is also reduced from 18.9 to 16.7 km using the nudging initialization. Although the assimilation of total HFR velocities further decreases the mean distance obtained when only using





GO, results are slightly degraded when using nudging compared to restarting directly from the analysis, with a mean distance of 14.0 km after 48 h. On the contrary, the assimilation of HFR radials benefits from the nudging approach in contrast with the direct restart, reducing the mean separation distance from 14.3 to 13.4 km after the first two days (which represents a 51% reduction in comparison to $CR$) and giving the best results among all simulations using this initialization method

## 4 Discussion

The assimilation of high-resolution HFR surface currents observations in a reduced part of the modelling domain could have a negative effect on the rest of the variables under the effect of spurious model error correlations. While in Stanev et al. (2015, 2016) the positive outcome of the data assimilation extends beyond the HFR covered area, Zhang et al. (2010) showed how, in their experiments, the assimilation of HFR led to an improvement of surface currents but a degradation of the sub-surface temperature forecasts. Sperrevik and Christensen (2015) evidenced that using TS profiles along with HFR observations led to better results, as they control the density fields while adding a constraint on the circulation. Here we show that the assimilation of local HFR (both totals and radials) observations along with the generic ones does not degrade the improvement on SLA, SST fields and on Argo TS profiles achieved over the whole domain when assimilating only the generic observations. The results obtained for all experiments with DA show similar performance in this sense. Differences mostly depend on the type of initialization employed, with small variations in the RMSD obtained for the Argo salinity. Nevertheless, this work is mainly focused on the study of surface currents and thus, the impact on sub-surface fields has not been deeply analyzed. CTD casts or glider data in the region should help to complete the assessment in future studies.

We have used DIVAnd reconstructed fields as a benchmark for our Lagrangian validation. These hourly fields properly represent the inertial oscillations, compared to other gap-filling techniques (Barth et al., 2021), and we consider it as the best possible high resolution observation of the surface currents in the area which allows the simulation of Lagrangian trajectories. It is very positive that the skill scores obtained for the HFR DA experiments are very close to that obtained by DIVAnd. While DIVAnd outperforms the capabilities of the WMOP DA system inside the coverage of both HFR antennas, it is the opposite outside this region, demonstrating the capacity of the model to improve the representation of the currents beyond the HFR coverage area. The assimilation of GO, in particular SLA, constrains the geostrophic circulation, leading to a better representation of the Balearic current and an increase of the $SS$ in that area. The importance of this constrain is highlighted when comparing with DIVAnd-derived trajectories, which do not properly represent these features. While the mean $SS$ for the DIVAnd-derived trajectories inside the area is 0.53, it drops to 0.29 outside of it, being significantly lower than all model-derived trajectories. This behaviour is consistent with Barth et al. (2021) which show that the DIVAnd reconstructed fields outside the area covered by both HFR antennas are much less reliable. Our results demonstrate the utility of dynamical models assimilating high-resolution observations as good alternatives to data-driven short-term forecasting methods, due to their capacity to extend the correction beyond the observation coverage area. They also show the importance of combining HFR and altimeters observations which help to constrain the geostrophic circulation.





Two different initialization strategies have been evaluated. While restarting directly from the analysis may introduce some

high frequency and spurious waves or instabilities in the system due to inconsistencies between the corrected fields and the model equations, it considers an initial state which is closer to observations. On the other hand, the nudging strategy provides a more conservative framework, in which the model dynamics are better respected but with the drawback that some of the correction achieved with the observations may be lost. In general, both approaches show similar results leading to a reduction of the RMSD over the whole domain. The use of the $nudging$ strategy also leads to an improvement of the predictions of

surface currents when adding HFR observations, compared to the simulation that only uses generic data sources. It is important to point out that, in our case, nudging is only applied towards the temperature, salinity and sea surface height fields, but not towards the velocity fields. Therefore, the assimilation of the surface currents enables to correct the density fields, which in turn improves the surface velocities due to the model initial adjustments.

The $nudging$ strategy limits the possible shocks and anomalous gradients that may be generated in the analysis and remains

closer to the physical balances. We found that it was not optimal for surface currents prediction when using HFR total velocities but a better choice for radial data. This is probably due to the fact that reconstructed total velocities are already smoothed out through a pre-processing step contrarily to radial data, which are more noisy and then directly benefit from the smoothing effect of the nudging approach. The nudging strategy appears to be a good solution for operational purposes, when the ocurrence of noisy data tends to be more frequent. It may also be a good choice for systems depending on operational data sources for which

HFR antennas, for instance, may not work during certain periods or satellite and Argo data may not be available on time. It could also be less sensitive to potential errors in data in cases where near real-time observations could have large errors.

The observation error is considered equal for total and radial currents in this study. It was the result of sensitivity tests after the evaluation on the effect of DA on both surface current corrections and vertical T-S profiles as represented by Argo floats. Some authors used different observation errors at a certain location depending on whether it is covered by a single

antenna or more than one (Vandenbulcke et al., 2017). Here we have considered the same error for all HFR radial observations independently of the antenna coverages, so as to also exploit the potential benefit of observations in areas covered by only one antenna, as discussed in Shulman and Paduan (2009); Stanev et al. (2015). In this sense, we believe that observations should not be considered less valuable depending on whether we have observations from other antennas at a certain location or not.

## 5   Conclusions

In this work, we have integrated different multivariate ocean observations with numerical modelling to improve the dynamical knowledge of ocean currents in line with the actual concerns in operational oceanography (De Mey-Frémaux et al., 2019; Kourafalou et al., 2015a, b; Schiller et al., 2015). We combined high-resolution modelling with satellite and in-situ observing sources and HFR surface currents measurements to discuss the contribution that the developing HFR networks (Roarty et al., 2019; Rubio et al., 2017) could provide to regional and coastal operational modelling. The impact of HFR-DA has been

evaluated, using both radial and total observations along with generic data sources as SLA, SST and Argo TS profiles. The



system showed its ability to improve the representation of ocean fields by assimilating different types of observations, from a variety of sources observing a wide range of spatio-temporal scales.

The assimilation of GO helps to correct surface currents in the IC as revealed by both the Eulerian and Lagrangian validations. The employment of HFR observations further improves the forecasting of surface currents in the IC. While $GNR$
simulations reduce the RMSD and the mean error, the assimilation of HFR leads to an increase in the correlation between model and observations for both components of the velocity. The $TOT$ experiment is the one that best fits the observations. Besides, it provides the best average skill score for Lagrangian prediction and the lowest mean separation distance between drifters and virtual particles. The use of radial observations benefits from the use of an intermediate nudging initialization approach after the analysis. The results presented in this study confirm the usefulness of HFR systems to improve regional
operational ocean forecasting models, even when providing a limited coverage with respect to the model domain extension.

*Data availability.* – Simulations are archived on the SOCIB server and are available upon request to info@socib.es.

– Drifter data can be accessed at https://doi.org/10.25704/mhbg-q265.

– HFR total currents can be accessed from https://doi.org/10.25704/17gs-2b59 and http://thredds.socib.es/.

– DIVAnd HFR reconstructed fields data and information can be found at https://doi.org/10.13155/78713.

– Argo data downloaded from IFREMER ( ftp.ifremer.fr)

– Sea Surface Temperature can be downloaded from NASA-JPL portal (https://podaac.jpl.nasa.gov/dataset/MUR-JPL-L4-GLOB-v4.1)

– Sea Level anomaly (SEALEVEL_EUR_PHY_L3_REP_OBSERVATIONS_008_061) and MED-MFC model (MEDSEA_MULTIYEAR_PHY_006_004) products can be downloaded from the Copernicus Marine Service (CMEMS).

*Author contributions.* JHL and BM conceptualized the experiments. JHL conducted the experiments, analysis and writing of this manuscript
with the support of BM and AO. ER helped in the interpretation of the HFR data and the discussion of the results. AS helped in the coding and in the discussion of the results. JT was in charge of SOCIB Observing & Forecasting System design and direction. All authors contributed to the review of the article.

*Competing interests.* The authors declare that they have no conflict of interest.

*Acknowledgements.* We want to thank all the data providers listed for making their data available. Also the Spanish Meteorological Agency
(AEMET) for providing the HIRLAM model outputs used. This work was partially funded by the EU Horizon 2020 JERICO-NEXT (grant agreement No 654410) and EuroSea (grant agreement No 862626) projects, as well as MEDCLIC, a joint project between SOCIB and "La



Caixa" Foundation. We would also like to thank Mélanie Juza, Eugenio Cutolo, Adèle Revelard, Eva Aguiar, Máximo Garcia-Jove, Sun Yong Kim and Alexander Barth for their technical support and fruitful discussions.



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
