# Peer review of "Evaluating High-Frequency radar data assimilation impact in coastal ocean operational modelling"

_Ocean Science, 2021_

## Author Response (AR1)

**REFEREE 1, Jeffrey Paduan**

I have reviewed the manuscript by Hernandez et al. entitled "Evaluating high-frequency radar data assimilation impact in coastal ocean operational modelling." As the title implies, this study is focused on high frequency (HF) radar impacts as a data source for ocean circulation models. HF radar networks have been proliferating in coastal areas worldwide for the primary purpose of mapping ocean surface currents out to ranges of 150 km with resolution of a few kilometers. The wide-area and real-time coverage derived from these networks makes them an important new tool for managing resources, responding to emergencies, and researching ecosystem responses in coastal areas. The goal of the present study and others before it is to develop and assess techniques for also using the HF radar-derived surface current mapping data as an assimilation source for 3-D circulation models. In this way, the impact of the HF radar observations can be expanded in both space and into the future through the use of models. The topic, therefore, should be of interest to a wide range of managers and scientists.

This manuscript is well written and the illustrations are clear. In addition, the authors provide a comprehensive and helpful review of past results involving the assimilation of HF radar-derived surface velocity. For these reasons, I recommend the manuscript for publication with only minor corrections.

The one substantive concern I have is to ensure that the basis for the conclusion that HF radar assimilation improves the model outside the range of the HF radar observations is accurate and consistent. It is mentioned in the abstract, in the final sentence of the conclusions, and in lines 344-346 that the data assimilation improves the results even outside the data coverage area. This conclusion may be true and it is strictly supported by the data, but I don't believe that the independent test data in this particular study is adequate to make the conclusion. In particular, Figure 9 shows that the results in the model domain outside the HF radar coverage area are actually better than they are within the HF radar coverage area. That result is dubious in terms of justifying the conclusion. It is more likely that the currents in the test region outside the HF radar coverage area happen to be very steady leading to a particularly good model-data match in that location, a possibility that is referred to in the text. Hence, I believe that the conclusion of improved data assimilation results outside the HF radar coverage area should be tempered or removed in this particular case.

I find Figure 10 to be particularly interesting and useful in terms of guiding operational systems. It illustrates (and quantifies) the clear benefits of data assimilation for traditional data sets and it shows the added benefit of assimilating HF radar data in addition to those traditional data sets.

MINOR COMMENTS:

Line 38: This sentence is awkward. Please review and revise it.

Line 111: "It operates since" should be "It has operated since"

Line 308 and Figure 8: The text refers to a set of 1000 Lagrangian particles but the figure refers to only 100 particles. Please reconcile this.

Line 406: "this constrain" should be "this constraint"

Line 427: "contrarily to" should be "contrary to the case for"

Dear Jeffrey,

Thank you for your time to review the manuscript. We very much appreciate your feedback and comments.

Regarding your main concern about the impact of HF radar data assimilation outside the range of HFR observations, we would like to clarify our results and conclusions as they may have been misleading in the

way they were expressed in the original manuscript. Our results (see Fig 9 and Table 3) show that: 1) the skill score is higher outside the area of HFR coverage for all simulations, 2) assimilating HFR data does not lead to any degradation of the model performance outside this coverage area with respect to GNR.

As you suggest the first point is related to the nature of the currents driving the drifters outside the HFR coverage area, which are more defined and steady and hence better described in the simulations, even with a limited number of assimilated data. The dependence of this result on the available dataset and specificities of the area are more clearly explained in the revised version of the manuscript. On the other hand, what we wanted to point out in the abstract and the conclusions is the fact that HFR DA helps to correct the currents inside the area covered by both antennas and that it does neither improve nor degrade them outside it in comparison to the GNR simulation. Our conclusions concerning the impact of HF radar data assimilation outside the coverage area have been clarified and tempered in that sense in the revised version of the manuscript.

The new text in the results part reads as follows:

*"...The SS of all simulations is higher outside the coverage area than inside due to the different nature of the currents, which are more defined and steady north of Ibiza Island and hence better described in the simulations. All data-assimilative simulations lead to a similar performance outside the coverage area (SS around 0.58), representing an improvement with respect to CR (SS=0.36). Although RAD obtains the best results (0.59), we believe that the actual validation dataset, given the steady circulation in the region, is insufficient to conclude if the HFR can improve or degrade the performance obtained in the GNR simulation outside the coverage area."*

The sentence in the abstract has been removed to avoid confusion and a sentence has been added in the conclusions to try to clarify this.

*"The Lagrangian validation reveals the capacity of HFR data assimilation to significantly improve the forecasting of surface currents inside the area covered by both antennas, while maintaining beyond this area the performance achieved by assimilating the generic observation sources."*

Thank you for the minor comment corrections. We have included them in the revised manuscript.

**REFEREE 2. Anonymous**

The paper describes the impact of assimilation of HF radar data at the Ibiza Channel (Western Mediterranean Sea). The authors assimilate commonly used data sets (Sea surface temperature, sea level anomaly and Argo profiles) in combination with HF radar data. For the HF radar data two options are considered: either assimilating the total currents (derived from the radial currents) or directly assimilating radial currents. Both assimilation experiments are validated against drifter observations. The authors conclude that the assimilation using total currents fits the Lagrangian observations the best.

**Comments:**

1. There are some general properties about the Kalman-based assimilation systems with transformed observations that should be mentioned to set the context of the study. If the hypotheses of the Kalman filter are verified (in particular the model is linear, error covariances are perfectly known), then the analysis would provide exactly the results under any invertible linear transformation of the observations (provided the observation operator and the obs. error covariance matrix are transformed accordingly).

The assimilation of any additional observation has the impact to reduce the error of the analysis on average. The consequence of these two properties is that the assimilation of transformed observations (possibly using

a non-invertible linear transformation) should not be better than the error using the non-transformed observations. In practice this can be shown by considering the observations associated to a zero singular value of the transformation and the observation with a non-zero singular value separately; the observations with a zero singular value are ignored by the transformation (these correspond to radial HF radar observations for which no matching second HF radar observation exist to derive total currents).

Under these, admittedly restrictive, assumptions, the assimilation of radial currents should work better than the assimilation of total currents. Intuitively, this makes sense because all the information of the total currents is already included in the radial currents and the radial currents have additional information not included in the total currents.

However, for real-world experiments there are some assumptions not verified which can lead to the opposite conclusion. In particular, we know that the model is non-linear, observation error covariances are not perfectly known and arbitrary observation operators cannot be specified by most current assimilation systems. Also it is not completely clear if the mapping from total currents to radial currents is a linear process (can you clarify this point?). I suggest that the authors include this additional information to clarify to the reader the motivation of this study.

2. The observational error covariance is a crucial parameter in the assimilation system, which is often not very well known because of the contribution of the representativity error.

Maybe I missed it but I did not see the particular values that were used. It is a bit surprising that the same error covariance values were used for radial currents and total currents. Can you expand this discussion by including the different values of the observational error covariance that were tested in your sensitivity test (line 432)? See also below.

I recommend the publication of this manuscript after revision.

Dear referee,

We would like to thank you for your time to carefully revise the manuscript and for your comments and feedback. The theoretical concerns about the use of Kalman-based assimilation systems with transformed observations are particularly interesting. As suggested, we have further developed this aspect in the introduction of the revised manuscript.

> *"Theoretically and under the assumptions of linearity and normal distribution of errors in the state dynamics and measurements, as well as in the transformation from radials to totals, the assimilation of radial currents should overperform the assimilation of total currents, since all the information of the totals is included in the radialsand the later contain additional information which is not included in the former. However, in real-world experiments, these major assumptions are not verified. In particular, the model is non-linear, observation error covariances are not Gaussian and certainly not perfectly known and the transformation from radials to totals also involves nonlinearities. In the literature, both kinds of observations have been assimilated with satisfactory results."*

The transformation from radial to total observations is non-linear and it can include different radial observations for each time step. The temporal threshold used for radial and total observations to perform the daily mean is also different (i.e. 25% for radials and 50% for totals), as we tried to clarify in the text (see below).

Concerning the second point, we would like to shortly explain the procedure followed to set the observation errors. First, we set a total observation error standard deviation of 0.1 m/s for HFR total observations, which accounts both for instrumental and representativity error. This value is consistent with local comparisons against surface currents measurements from a point-wise currentmeter (1.5 m depth) and a downward-looking ADCP (first bin at 5 m depth) carried out by Lana et al., (2016), which reported a RMSD between 0.07 and 0.12 m/s. This value of 0.1m/s was fixed in our experiments as it yielded to a proper correction of surface currents, without degrading the vertical structure.

Once the observation error was set for HFR total observations, we performed new experiments to evaluate the potential differences between the total and radial observations. Total observations were interpolated and projected to generate synthetic radial observations containing the exact same information as the totals but with a radial-like pattern in the area covered by both antennas.

The assimilation of these total and synthetic radial observations using the same observation error led to almost identical results in surface fields and vertical structure, with complex correlation of 0.92 and a RMSD of 0.02 m/s obtained between both analysis fields in the HFR grid points. Based on these results we decided to use the same observation error for both types of observations.

The last paragraph of the discussion has been expanded to include this explanation.

**Minor comments:**

Line 130: This is a bit confusing. Maybe you can expand this part: "It sometimes happens that there are enough radial observations to compute the total observation for most of the periods but with none of those radial observations satisfying the temporal threshold by itself."

The phrase has been reformulated as follows:

> *"Daily means of radials and totals are computed independently for each data type from the hourly observations. For the total currents the daily mean is only considered at grid points for which at least 50% of hourly measurements are both available and flagged as good, as also used by Lorente et al. 2015. In the case of the radials, a threshold of 25% is considered for computing the daily mean. As stated above, at each grid point, the hourly total currents are calculated using all available radial observations within a radius of 6 km. Consequently, some total observations could be computed using different radial grid points within this radius for each hour that individually do not satisfy the threshold of 50% imposed for the total velocities. Therefore, using the same threshold to calculate the daily means of both observations could lead to patches with available reconstructed daily mean total currents but no daily mean radials available. This is the reson why we decided to use a less restrictive threshold to have better radial spatial coverage, consistent with that of the total observations in the area covered by both antennas. "*

Equation 3: The notation is a bit odd as you have a vector on the left hand side and a scalar on the right hand side.

This has been corrected. Vectors of zonal and meridional velocity observations and angles are highlighted in bold.

Line 202: Notice that the nudging is not applied to the velocity fields: quite surprising. Did you also test nudging the velocity field?

No, we did not test nudging the velocity field. Nudging velocity fields can be problematic since it can significantly perturb the model balance equations. This is the reason why we decided to do it only towards T, S and SSH.

Indeed, we think that obtaining such a degree of correction on surface currents when applying the nudging only towards T, S and SSH is a relevant result. It means that surface current assimilation can correct T, S and SSH fields that, after ingestion and adjustments by the model dynamics, in turn improves the representation of surface currents. This point is specifically discussed in the third paragraph of the discussion.

Equation 5: ss: should it be upper-case SS?

Thanks. It has been changed for coherence with the rest of the manuscript.

Table 2, Table 4: can you also include the RMS (without normalization)? Can you also include in this table a validation metric which is sensitive to the direction of the current, not only the speed of the current? (e.g. the RMS error of u and v components individually?)

It has been included in both Tables 2 and 4.

Line 305: diffusion term: how large is the diffusion coefficient? And how was it determined?

The diffusion coefficient we used is 50m$^2$/s. This value was determined empirically based on the virtual particle dispersion after a few days in comparison to the dispersion of available real drifters in the Balearic Sea. It was successfully used in other studies with the WMOP model (Cabanellas Reboredo et al. 2019, Ruiz-Orejon et al. 2019, Compa et al., 2020, Kersting et al., 2020).

In the present study, we use the center of mass of 1000 particles to calculate the mean separation distance. We have verified that the results are not significantly affected by the value of the diffusion coefficient, which has a significant impact on the spread of the trajectories but not on the path of the mean trajectory

Following the reviewer's advice, this has been included in the new version of the manuscript as:

> *"After each advection step the diffusion is imposed using a random distribution with a diffusion coefficient of 50 m$^2$/s, in line with recent Lagrangian studies using this model (*Cabanellas Reboredo et al. 2019, Ruiz-Orejon et al. 2019, Compa et al., 2020, Kersting et al., 2020*). We have verified that the results are not significantly affected by the value of the diffusion coefficient, which has a significant impact on the spread of the trajectories but not on the path of the mean trajectory."*

Line 432: "The observation error is considered equal for total and radial currents in this study. ..." This is quite surprising as one would expect the radials a bit noisier and the total currents error variance should depend on the location (among others due to GDOP). Can the paragraph be expanded? Can you also include the value of the observational error covariance?

As stated above, the choice of the observation error variance for radial observations was the result of model experiments simulating radials from a given set of total observations. The dependence of the error with the location or the availability of either one or two antennas is discussed in the manuscript. We agree that the representation of the observation error could be refined in our system, also by including correlated

observation errors, even if our knowledge of these errors is still somehow limited. It is an interesting aspect that should be evaluated in future studies.